

# Estimating chemical composition of atmospheric deposition fluxes from mineral insoluble particles deposition collected in the Western Mediterranean region

Yinghe Fu[1], Karine Desboeufs[1], Julie Vincent[1], Elisabeth Bon Nguyen[1], Benoit Laurent[1], Remi Losno[1+], François Dulac[2]

[1]Laboratoire Interuniversitaire des Systèmes Atmosphériques (LISA), UMR7583 CNRS, Université Paris 7 Denis Diderot, Université Paris-Est Créteil, Institut Pierre-Simon Laplace, Créteil, 94010, France
[+] Now at Institut de Physique du Globe de Paris
[2]Laboratoire des Sciences du Climat et de l'Environnement (LSCE), UMR 8212 CEA-CNRS-UVSQ, Institut Pierre-Simon Laplace, Gif-sur-Yvette, 91190, France

*Correspondence to*: Yinghe. Fu (Yinghe.fu@lisa.u-pec.fr)

**Abstract.** In order to measure the mass flux of atmospheric insoluble deposition and to constrain regional models dust simulation, a network of automatic deposition collectors (CARAGA) has been installed throughout the western Mediterranean basin. Weekly samples of the insoluble fraction of total atmospheric deposition were collected concurrently on filters at 5 sites including 4 on western Mediterranean islands (Frioul and Corsica, France, Mallorca, Spain, and Lampedusa, Italy), and 1 in the southern French Alps (Le Casset), and a weighing and ignition protocol was applied in order to quantify their mineral fraction. Atmospheric deposition is both a strong source of nutrients and metals for marine ecosystems in this area. However, there is little data on trace metal deposition in the literature since their deposition measurement is difficult to perform. In order to obtain more information from CARAGA atmospheric deposition samples, this study aimed at testing their relevance to estimate elemental fluxes in addition to total fluxes. The elemental chemical analysis of ashed CARAGA filter samples was based on an acid digestion and an elemental analysis by inductively coupled plasma atomic emission spectroscopy (ICP-AES) and mass spectrometry (MS) in a clean room. The sampling and analytical protocols were tested to determine the elemental composition for mineral dust tracers (Al, Ca, K, Mg, and Ti), nutrients (P and Fe), and trace metals (Cd, Co, Cr, Cu, Mn, Ni, V and Zn) from simulated wet deposition of dust analogues and traffic soot. The relative mass loss by dissolution in wet deposition was lower than 1 % for Al and Fe, and reached 13 % for P due to its larger solubility in water. For trace metals, this loss represented less than 3 % of the total mass concentration, except for Zn, Cu and Mn for which it could reach 10 %, especially in traffic soot. The chemical contamination during analysis was negligible for all the elements except for Cd which is in very low concentration in dust. Tests allowed us to conclude that the CARAGA samples could be used to estimate contents of nutrients and trace metals in the limits of loss by dissolution. Chemical characterization of CARAGA deposition samples corresponding to the most intense dust deposition events recorded between 2011 and 2013 has been performed and showed elemental mass ratios consistent with the ones found in the literature for Saharan dust. However, the chemical analysis of CARAGA samples revealed the presence of some anthropogenic signatures, as for instance high Zn concentrations in some



samples in Lampedusa, and also pointed out that mineral dust can be mixed with anthropogenic compounds in the deposition samples collected on the Frioul Island. Results showed that the chemical analysis of CARAGA ashed samples can be used to trace back origins of elemental deposition. The elemental atmospheric fluxes estimated from these chemical analyses of samples from the CARAGA network of weekly deposition monitoring constitute the first assessment of mass deposition fluxes

of trace metals and P during intense dust deposition events at the scale of the western Mediterranean basin.

## 1 Introduction

The Mediterranean basin is a receptor for the deposition of particles emitted in surrounding coastal urban areas and continents. These deposited particles can have both natural (e.g. Saharan dust and biogenic emissions) and anthropogenic origins (e.g., Loÿe-Pilot and Martin, 1996; Kanakidou et al., 2011). Atmospheric inputs supply as much nutrients as riverine inputs in the

Mediterranean and are less confined to coastal waters (e.g. Moon et al. 2016; Richon et al., 2017). They play a significant role in marine nutrient cycles during the Mediterranean water stratification period, i.e. from May to September. This is the case for P and N macronutrients (Loÿe-Pilot et al., 1990, Pulido-Villena et al., 2010) and Fe micronutrients (Bonnet and Guieu, 2006). Saharan dust deposition is also considered as an input of trace metals, such as Co, Ni, Mo, Mn, Zn and Cd, for the Mediterranean Sea, which play an essential role in phytoplanktonic activity (Morel and Hudson, 1985). For instance, $N_2$

fixation and growth of diazotrophs could be controlled by dust inputs of trace metals (Ridame et al., 2011).

The ADIOS program, based on a network of deposition measurements with a common protocol deployed during the same period at nine stations throughout the basin, provides key information for studying spatial variability of nutrients and trace metals deposited in the Mediterranean area, but it was limited to only one year (June 2001-May 2002) with a monthly resolution that cannot isolate high deposition events (Guieu et al., 2010). The measured elemental fluxes show a high spatial variability

and suggest that the dust-derived Fe deposition is higher in the Northwestern basin. For P deposition, the measurements seem to indicate a longitudinal gradient with significantly lower values in the eastern Mediterranean basin attributed equally to dust and anthropogenic inputs. However, due to the high temporal variability of desert dust events transported over the Mediterranean basin, it is difficult to conclude much from a one-year data set of the chemical elements associated with dust. To do that, a long-term and large-scale deposition sampling network is required (Schulz et al., 2012). In order to obtain a long

time-series of deposition mass fluxes at several sampling sites, a CARAGA collector (Collecteur Automatique de Retombées Atmosphériques insolubles à Grande Autonomie) has been developed (Laurent et al., 2015) and implemented at 5 sites in the western Mediterranean region (Vincent et al., 2016). This collector has up to 6 months autonomy without human intervention. To do that, only the insoluble fraction of total deposition is sampled on filters, excluding the soluble fraction of the deposition for which the preservation over months can be difficult.

Samples of atmospheric deposition are automatically collected by CARAGA collectors since 2011 on 5 sites: le Casset (S-E France), Corsica Island (France), Frioul Island (France), Mallorca Island (Spain), and Lampedusa Island (Italy). Measured total mass fluxes of atmospheric deposition and the weighing and ignition protocols used to quantify the mineral deposition





are presented in Laurent et al. (2015) and Vincent et al. (2016). In the present study, we investigate the possibility of using the CARAGA ashed samples to estimate the elemental mass deposition flux of nutrients and trace metals. Firstly, we tested with aerosol analogues the elemental composition changes during the sampling and treatment protocol of deposition samples. Secondly, we estimated the nutrients and trace metals fluxes of the most intense dust deposition events associated to dust

events over western Mediterranean basin between 2011 and 2013.

## 2 Materials and methods

### 2.1 Sampling of insoluble deposition and total mass measurements

Weekly deposition samples were collected between 2011 and 2013 with CARAGA collectors at 5 stations in the western Mediterranean basin presented in Fig. 1: Le Casset (44°59 N, 6°28 E, S-E France at 1850 m in altitude, rural area, ~ 170 Km

from sea shore), Corsica Island (43°00 N, 9°21 E, France, at 75 m in altitude, 300 m from sea shore), Frioul Island (43°15 N, 5°17 E, France, at 45 m in altitude, in front of Marseille (city of 1. 57 million population according to INSEE 2012), Mallorca Island (39°15 N, 3°03 E, Spain, at 7 m in altitude, 70 m from sea shore), Lampedusa Island (33°21 N, 10°30 E, Italy, at 38 m in altitude, ~ 20 m from the north-western coast of Lampedusa), more sampling sites information can be found in Vincent et al. (2016). The sites positions were selected to cover the western basin by integrating East-West and North-South gradients.

In the CARAGA collectors, dry and wet deposition is collected by a funnel (0.2 m$^2$) during one week starting on Thursday noon. At the end of the week, the funnel is rinsed by 250 mL of ultrapure water. Rinse water and rain water pass through an AA Millipore® cellulose ester filter with a 0.8 μm porosity in order to collect the insoluble fraction of the deposition. A complete description of the CARAGA collector and sampling network can be found in Laurent et al. (2015) and Vincent et al. (2016). After being brought back to laboratory, collected filters were ashed following a progressive increase in temperature up

to 550°C in order to destroy and vaporize the filter and organic matters. The mineral deposition mass, excluding volatile and organic matter, was estimated by weighing the ashed samples (Laurent et al., 2015), and the ashes were stored in acid-cleaned Eppendorf tubes.

From 2011 to 2013, 108 samples corresponding to the most intense weekly dust deposition samples were studied by Vincent et al. (2016), and Saharan origin was identified for 107 samples selected samples by using satellite data and air mass

trajectories. The selected samples accounts in mass for 84 %, 78 % and 73 % of the total deposition in Lampedusa (37 samples), Mallorca (21 samples) and Corsica (11 samples), and it contributes for around 50 % in Frioul (21 samples) and Le Casset (17 samples). A complete description of treatment and selection of samples can be found in Vincent et al. (2016).

### 2.2 Analytical protocol

In order to further obtain their elemental composition, ashed CARAGA samples were digested using a digestion protocol

adapted from Heimburger et al. (2013). About 20 mg of ashed samples was weighed and transferred to a Savillex™ PFA digestion vessels with 0.5 ml of ultrapure water to avoid the loss of sample. The samples were then digested using 2.5 mL of



mixture of Suprapur® acid (1 mL HCl 30 %, 1 mL HNO$_3$ 65 %, and 0.5 mL HF 40 %) during 14 h in an air oven at 130 °C in closed vessels. When the digestion vessels were cooled, acids were completely evaporated on a heater plate. Overall, 130 µL of suprapur® HNO$_3$ and then 13 mL ultrapure water were added. Finally, the content of each vessel was transferred into an acid-cleaned 15 mL tube for elemental analysis. All the protocol of analysis was made in ISO class 5 clean room with material

washed with Ultrapur then Suprapur® HCl. At least 3 blanks (ashed filters without particles) were produced for each set of acid digestion in order to estimate the potential contamination during protocol.

Analyses were performed by a SPECTRO ARCOS (http://www.spectro.com) inductively coupled plasma-atomic emission spectrometry (ICP-AES) for markers of desert dust (Al, Ca, K, Mg, Na and Ti), nutrients (P and Fe), and traces metals (Cd, Co, Cr, Cu, Mn, Ni, V and Zn). Due to digestion with hydrofluoric acid, and the high volatility of formed SiF$_4$, we could not

get the contents of silicon.

**2.3 Test of elemental mass loss during the CARAGA protocol**

In order to estimate the change in elemental composition during the CARAGA sampling and treatment protocol, tests have been carried on analogues of typical Saharan dust or soot particles, which were subject to the collection protocol and then ignition in the laboratory.

Wet deposition being the major pathway to solubilize nutrients and trace metals, analogues of typical wet deposition CARAGA samples were made by simulating rainwaters containing two kinds of particles: dust and soot, which are the main insoluble aerosols (Ault et al., 2011). Desert dust samples were prepared from the fine fraction of soils (< 20 µm) collected in semi-arid areas of Douz (Tunisia) and Banizoumbou (Niger). The elemental solubility in desert dust varies depending on several factors, the mineralogical composition of the particles being one of the major ones (Journet et al., 2008). We selected these soils

because their mineralogical and elemental compositions were well known (Guieu et al., 2010; Paris et al., 2011; Desboeufs et al., 2014). In addition, they represent 2 extreme mineralogical and chemical characteristics observed in the dust emission zones of northern Africa influencing the western Mediterranean atmosphere (Bergametti et al., 1989): the Douz soil is rich in calcium carbonate and clay (Ca/Al = 4.16 and Fe/Al = 0.51) and the Banizoumbou soil is rich in iron oxides and quartz (Ca/Al = 0.06 and Fe/Al = 0.63) (Formenti et al., 2014). Soot samples are traffic particles collected in a tunnel in Paris. Indeed, black carbon

measured in the western Mediterranean is mainly anthropogenic carbon-containing particles from combustion emission (Mallet et al., 2016).

Simulated rainwaters were made by adding 50 ± 0.2 mg of particles (fine fractions of soils or soot) into 250 ± 2 mL of ultrapure water acidified at pH = 4.7 with Suprapur® sulfuric acid. This pH value and the sulfuric acid are characteristics of atmospheric conditions and are commonly used to mimic the release of trace metals from aerosol in rainwater (Desboeufs et al., 2001).

After 30 min of contact time with automatic shaking, we reproduced the CARAGA sampling of wet deposition through filters and collected the filtrated solution. The time of 30 min corresponds to the most prevalent average time of rainfall in the western Mediterranean basin (Rysman et al., 2013). 24h later, we rinsed the filter holder using 100 ± 5 mL of ultrapure water for simulating the rinsing step of the CARAGA protocol. We did 3 replicates for each type of particles. The filtrated rinsing water




was collected. Dissolved nutrients and trace metals concentrations were determined in the filtrated simulated rainwaters and rinsing waters to estimate the loss associated to the soluble fraction during the collection of the insoluble fraction. The dissolved concentrations were analyzed by ICP-AES for major elements: Al, Ca, P, Fe and Ti, and by a Thermo Fisher Scientific™ Element 2, high-resolution inductively coupled plasma mass spectrometry (HR-ICP-MS) for trace metals Cd, Co, Cr, Cu, Mn, Ni, V, and Zn. Rainwater blank values corresponding to acidified ultrapure water without soils passed through the collector (n=3) were subtracted to the measured concentrations.

In order to assess the potential change of the sample elemental composition due to the calcination step, we simulated sampling of rainwaters using the Douz and Banizoumbou soils with the above described protocol but by adding 90 mg of soils. After filtration and rinsing, filters were dried in the oven at 40°C for 2 h and then weighed, about 20 mg dry soils were collected in a clean Eppendorf tube, the rest (with filter) was ashed based on the CARAGA protocol, then weighed, and about 20 mg were also collected in an Eppendorf tube. Both of them were digested for analysis by ICP-AES and compared. This test was made in triplicates for each soil.

## 2.4 Estimation of contamination during CARAGA protocol

107 selected samples were digested and analyzed for markers, nutrients and trace metals using the proposed protocol. In order to estimate the contamination and limit of detection (LoD) of the analytical protocol, 9 blanks of CARAGA samples (filters exposed but not having collected) were used and processed. Even if the protocol of chemical analysis was tested for 20 mg of ashed samples, the mass of several CARAGA samples was inferior to 20 mg, especially the less intense deposition samples. In this case, the totality of ashed samples was used for acid digestion. The minimum sample mass was 2 mg. The weighing uncertainty was ± 0.2 mg. It was not significant (1 %) for samples of 20 mg, but for samples of 2 mg, it could represent 10 % of uncertainty. However, these samples made a low contribution to total annual mass fluxes (Vincent et al., 2016). So, we considered uncertainty in weighing protocol was not significant for annual mass flux of nutrients and trace metals. We compared the lowest concentrations obtained for a 2 mg sample with blanks to check the reliability of measured concentrations (Table 1).

For P, Cr, Cu and Zn, concentrations of blank values were below the limit of detection, so we considered that contamination could be neglected. For others elements except Cd, blank concentrations were quantifiable and presented low variability but remained largely inferior to minimum concentrations in 2 mg CARAGA samples. Cd concentration was inferior to LoD, so we could not estimate Cd content in samples whose mass is low. For mass fluxes calculation, we subtracted these blanks values to elemental concentrations obtained in CARAGA samples. We did not take into account Cd in the following study because of the large uncertainty on its concentrations.



## 3 Results and discussion

### 3.1 Estimate of elemental mass loss during the CARAGA protocol

Averages of elemental mass loss and the standard deviation of 3 replicates for 3 types of particles (soils of Douz and Banizoumbou, and traffic soot) during the sampling of wet deposition are presented in Table 2. Loss by dissolution during

collection and rinsing has been calculated as fractional solubility i.e.:

$$S = \frac{\text{Elemental mass in filtered and rinsed solution}}{\text{Elemental mass in initial particles}} \times 100 \text{ \%}.$$

The largest mass losses were observed for major alkali and alkaline earth metals: Ca, K, Mg and Na. Al and Ti which are the major tracers of dust presented solubility lower than 1 % for the two dust analogues. The low solubility of Al and Ti makes possible of using them as markers of sources in CARAGA samples, but it is not the case for the ratio K/Al and Ca/Al typically

used to characterize the desert dust sources (Scheuvens et al., 2013). For nutrients, the mass loss for Fe (0.11-0.68 %) was lower than 1 % for all types of aerosols analogues, whereas the maximum mass loss of P reached $13.3 \pm 2.3$ % in soils of Douz. The results on the fractional solubility were in agreement with the background about P and Fe solubility in mineral dust (Anderson et al., 2010) and in black carbon particles (Desboeufs et al., 2005). For trace metals, the mass loss was typically under 3 % except for Zn (4.8-8.8 %), Cu (1.7-6.6 %) and Mn (2.8-10.8 %) for which the solubility was lower than 10 %. In

the literature, few data are available on the aerosol fractional solubility in rainwater and in particular for dusty rain, solubility values of Al (0.24 %), Fe (0.06 %), Cu (8.37 %) and Mn (5.62 %) in rainwater in Istanbul reported by Başak and Alagha. (2004) are consistent with this study. The solubility values are variable from an aerosol analogue to the other: Mn present in the Banizoumbou soil is more soluble than in traffic soot, whereas Cu, Co, Ni, Cd and Zn contained in traffic soot are more soluble than the ones in Banizoumbou soil. The ICP-AES analysis performed on samples before and after the ignition protocol

point out that elemental mass loss during the calcination were not significant for tracer elements (Al and Ti), nutrients (P and Fe), and trace metals (Co, Cr, Cu, Mn, Ni, V and Zn).

The tests emphasized that the main change in chemical composition between deposited aerosols and ashed CARAGA samples were due to the loss by dissolution during wet deposition and by rinsing. The composition of nutrient and trace metals obtained from simulated CARAGA samples were representative of initial dust and soot analogues with a maximum of 13 % of

underestimation for P, less than 10 % for Zn, Cu and Mn, and less than 5 % for other elements. Tests were carried out for wet deposition. In the case of dry deposition, particles are rinsed and the contact time with water is less than for wet deposition. Thus, the underestimation due to dissolution would probably be lower in the case of dry deposition. The highest uncertainty of 13 % is inferior to the common discrepancy on the deposition fluxes estimated from different dry deposition samplers (Goosens and Rajot., 2008; Lopez-Garcia et al., 2013). To conclude, taking into account the underestimate of soluble fraction,

the CARAGA samples seem to be relevant for estimating the total deposition fluxes of Al, Fe, P, Co, Cr, Cu, Ni, Mn, Ti, V and Zn.



## 3.2 Chemical characterization of CARAGA samples

After samples analysis, weekly elemental mass fluxes for the 107 deposition samples were calculated from measured concentrations. These samples being associated to deposition of Saharan dust, we checked their chemical compositions with Principal Component Analysis (PCA) and mass ratios X/Al, then compared with Saharan dust composition in order to validate

the use of the CARAGA samples to determine the origin of elemental deposition fluxes.

### 3.2.1 Principal Component Analysis (PCA) for 107 CARAGA samples

Principal component analysis was used to identify the possible sources of the 107 intense events deposition samples. Correlation between variables and factors for elements: Al, Fe, P, Ti, Co, Cr, Cu, Mn, Ni, V and Zn, and commutative variance for 5 sampling sites were presented in Table 3. For each site, 2-4 possible sources were identified, and the first source

contributed always much more than half the variance of data, and the two first factors were extracted as principal components that explained > 82 % of the variance of the samples data. The lower variances of first factor explained 57.34 % and 66.87 % at sites Lampedusa and Frioul respectively, showing more influence of other sources at these 2 sampling sites. In the studied 107 samples, all elements were globally issued from a similar source for each site, but other sources (especially of Zn, Cu, Cr, Co and P) influenced the sampling sites too, especially at sites Lampedusa and Frioul. In order to identify the outlier samples,

individual observations located on plane factors 1, 2 or 1, 3 are presented in Fig. 2, and the observations located higher (> 1 %) on axis Y (factor 2 or 3) were considered as outlier samples responsible to the bad correlation between variables and the first factors for studied elements.

### 3.2.2 Characterization of CARAGA samples by mass ratios

In order to identify the main source of each site, Al contents and mass ratios X/Al were assessed because Al is a typical marker

of dust particles. The mean Al contents in the CARAGA samples for the different sites was estimated by plotting total mass fluxes as a function of Al mass fluxes. Average of Al contents, standard deviation and the determination coefficient ($R^2$) for every site are presented in Table 4. For the 4 stations of Corsica, Lampedusa, Le Casset and Mallorca, $R^2 \geq 0.80$ and Al contents were typical of Saharan dust particles (Formenti et al., 2014). These values are also in agreement with the results of Guieu et al. (2002) on Al concentration in wet dust deposition ranging between 6.0 and 8.3 %. This indicates that African dust

is the main contributor to the mass deposition fluxes, consistent with the conclusion of Vincent et al, (2016), that validated sampling and analytical protocols. No clear correlation between Al concentrations and mass fluxes can be found in Frioul ($R^2 = 0.35$), pointing out that the deposited mass could be influenced both by African dust and other aerosols particles.

It is also interesting to note that Al contents increased with a South to North gradient, i.e. with the lowest values at Lampedusa, the closest site to Saharan dust sources and the highest values at Le Casset, the most distant site from Sahara (Table 4). Vincent

et al. (2016) emphasized that most of the time the origin of the deposited dust changed from one sampling site to the other: dust samples collected at Le Casset were mainly transported from the western part of the Sahara whereas dust arriving to





Lampedusa generally came from Tunisia and Lybia. Nevertheless, the values of the Al content were probably not only related to the provenance of dust since the emission sources providing dust deposition in Corsica and in Mallorca were generally the same (Western Sahara/southern Morocco and Tunisia/eastern Algeria) (Vincent et al., 2015) and yet the Al content was different. Desboeufs et al. (2014) observed an increase on the Al content in dust during its settling in the water column after

deposition to the ocean. This increase was explained by the fact that the mass loss due to dissolution of highest soluble species during settling modifies the mass percentage of the less soluble elements. The same effect could be observed in our samples due to the dissolution of soluble species during wet deposition as Ca, whose solubility can be increased by conversion of calcite to Ca sulfate or Ca nitrate during long range transport (Scheuvens et al., 2013). Vincent et al. (2016) showed that 80 %, 63 %, 60 % and 53 % of studied samples corresponded to wet deposition events at Le Casset, Corsica, Lampedusa and Mallorca,

respectively. The increasing contribution of wet deposition with the distance to the dust source is consistent with the obtained Al contents, since the farther from source the site is, the more important the number of wet deposition samples and the higher Al content. The consistency of Al content values with the dust references confirms the Saharan origin of deposition and supports the reliability of CARAGA data. However, the variability of the soluble part with the distance makes difficult to use CARAGA samples to estimate the elemental content of deposition samples and hence to directly compare the composition of

the deposition at the different sites. In the literature, inter-elemental mass ratios X/Al are often used to characterize the dust chemical composition and source (e.g. Formenti et al., 2008). Furthermore, Al has a low solubility in water, and trace metals solubility is also relatively low in dust (< 5 % in Soil of Douz and < 11 % in soil of Banizoumbou), so that such ratios should not be strongly affected by dissolution and hence can be used to validate the use of CARAGA samples to estimate the origin of atmospheric elemental fluxes.

The average of inter-elemental mass ratios X/Al, standard deviation (SD) and the determination coefficient ($R^2$) for every site, and some references values are presented in Table 5. A correlation of $R^2 > 0.65$ was obtained between trace metals and Al contents whatever the sampling sites and the X/Al ratio were close to values found in the literature for Saharan dust, except for Co, Cr , Cu or Zn at Lampedusa, Frioul and to a lesser extent at Le Casset.

At Lampedusa, the correlation are relatively low ($R^2 < 0.6$) for Fe, Cr, Cu, Mn and Zn. But excluding the 5 outlier samples

(identified by PCA in Fig. 2f) (column Lampedusa* in the Table 5), we obtained better correlations with $R^2 > 0.7$ and good consistence with reference dust values, except for Zn, the average of mass ratio ($35.08 \times 10^{-3}$) was 20 times larger than references values ($1.50$-$1.70 \times 10^{-3}$), that means that there was a very strong and frequent Zn anthropogenic source affecting deposition in Lampedusa during the sampling period. Anthropogenic Zn from manufacturing of non-ferrous metal and from incineration of wastes in the Mediterranean environment were reported by Ridame et al. (1999) and Guieu et al. (2010). In consequence, the

atmospheric fluxes of Zn in Lampedusa could not be considered as representative of dust deposition at this site. For the other trace metals, we can conclude that the main contribution of fluxes could be attributed to dust deposition during intense dust deposition event.

At Frioul, we already observed the low correlation of Al mass with total mass showing a contribution of other sources of particles to the deposited mass. Indeed, Frioul Island is a rock/sand island in front of the industrialized city Marseille, which





could be affected by resuspended mineral particles or by local anthropogenic sources. This conclusion is confirmed by the values of correlation found between Al and trace metals, as Cr, Cu or Zn ($R^2$ between 0.01 and 0.24), which were in general linked to one outlier sample (Fig. 2d) for Zn and Cu (31.3 % and 64.5 % in mass respectively), and 4 outlier samples (3 points in Fig. 2d, 2e identified by PCA and another one identified by mass ratio Cr/Al) for Cr (46.6 % in mass), However, even after

5 removing the outlier points, the median mass ratios Cr/Al (2.34 $10^{-3}$), Cu/Al (1.24 $10^{-3}$) and Zn/Al (11.29 $10^{-3}$) are larger than dust reference values (Table 5). The Zn/Al values were in agreement with the previous values obtained in the north-western Mediterranean (3-48 $10^{-3}$) where a very high anthropogenic component was found for Zn (Guieu et al., 2010), and according to Table 3, Zn and Cu had same origin. For this site, the interpretation of the chemical results is tricky since even during intense dust events we suspect a mixing with other anthropogenic atmospheric compounds that can be not important in terms of mass

but impact the chemical dust signature.

At Le Casset, Cu and Zn present correlations with Al with $R^2 = 0.26$ and 0.49. This correlation is due to 4 outlier samples (identified in Fig. 2b), after excluding these samples influenced by anthropogenic sources, which present a low proportion in mass (14.9 % and 10.7 % for Cu and Zn respectively), the correlation coefficients are higher than 0.65 and the obtained ratios are consistent with reference values for Saharan dust (Cu: 0.43 $10^{-3}$ (0.38-0.50 $10^{-3}$) and Zn: 1.60 $10^{-3}$ (1.01-1.70 $10^{-3}$) in Table

5). According to Table 3, a P source (factor 4) influenced the site Le Casset, because of the agricultural contribution in this rural location, and 2 outlier samples were responsible of the bad correlation of Co according to Fig. 2a. Thus, at Le Casset, the contribution of outlier samples lower than 15 % for elemental mass fluxes and the good agreement between measured and referenced ratio confirm the main dust origin of Co, Cr, Cu, Fe, Ti, Mn, Ni, V and Zn in deposition samples. In Corsica, the lowest correlation with Al obtained for Cr is due 2 outlier samples (in Fig. 2c) presenting 36.9 % in Cr deposited mass. Thus,

at this site, the Cr fluxes correspond to a mix between dust and other sources. Finally, at Mallorca, the correlations higher than 0.6 (and even 0.9 excluding Co) show that the metals fluxes are typically associated to dust deposition.

A high correlation of P and Al is obtained at Mallorca ($R^2 = 0.97$) and Corsica ($R^2 = 0.84$) with a P/Al ratio (0.010-0.012) in agreement with typical dust composition (Table 5). We conclude that dust deposition was the main provider of P during intense dust event deposition at these sites. On the contrary, the nonlinear trend obtained between P and Al fluxes at Le Frioul, Casset

and Lampedusa shows that the P fluxes measured at these sites cannot be used for estimating P fluxes associated with dust deposition. Main parameters affecting P contents in atmospheric deposition are its fractional solubility (precipitation and duration of rain) or anthropogenic sources. To check the effect of dissolution, these ratios were calculated for dry and wet deposition separately, but no linear trend was obtained for dry nor for wet deposition samples. Concerning the anthropogenic influence, the nonlinear trend is mainly due to a large repartition of data. It is therefore difficult distinguishing the fraction of

P from Saharan dust and anthropogenic or other sources at these 3 sites.

To conclude on the validation of CARAGA samples to estimate origin of atmospheric deposition, the results showed that the atmospheric deposition of trace metals at Le Casset, Corsica, Mallorca and Lampedusa is mainly associated to dust fluxes during intense dust event, except Zn in Lampedusa and Cr in Corsica. The outlier samples at these sites represented less than 10 % of deposited elemental mass. Thus the CARAGA samples can be used to estimate the atmospheric fluxes of these element



issued from dust deposition. On the contrary, the trace metals composition of CARAGA samples of Frioul was influenced by anthropogenic sources and the atmospheric fluxes are not representative of dust deposition even during intense dust events. This is also the case of P fluxes at the sites Le Casset, Frioul and Lampedusa, whereas P fluxes in Mallorca and Corsica can be used to estimate the dust-derived P deposition fluxes.

## 3.3. Weekly mass fluxes of nutrients and trace metals

The weekly mass fluxes of P and trace metals and percentage of wet deposition (wet only and mixed deposition according Vincent et al. (2016)), estimated for 107 samples covering the 2011-2013 sampling period, are presented in Table 6. The trace metals fluxes in Table 6 are presented in 2 categories: the fluxes mainly associated to dust deposition (sites of Le Casset, Corsica (except Cr), Mallorca and Lampedusa*), and the fluxes associated to deposition of a mixing between dust and other sources, as local or anthropogenic sources (sites of Lampedusa and Frioul). As observed by Vincent et al. (2016) on total dust mass fluxes, a north to south increasing gradient on Al and trace metals inputs linked to dust deposition is observed with the lowest mean weekly mass fluxes on Le Casset and the highest for Mallorca. Even if the island of Lampedusa is the closest site of African sources and the annual mass fluxes in Lampedusa are more important than in Mallorca (Vincent et al., 2016), the recorded weekly metals-bearing dust fluxes on this site (Lampedusa* in Table 6) remains inferior to the one recorded at Mallorca. Mass fluxes of P and trace metals linked to wet deposition are predominant at Le Casset, Corsica, Mallorca and Frioul (Table 6). In Lampedusa, it appears that wet deposition predominates for dust deposition events (Lampedusa*) whereas the metal fluxes including anthropogenic sources is mainly as dry deposition. The lowest dust metals fluxes found in Lampedusa* in comparison to Mallorca can be explained by the highest contribution of wet deposition in Mallorca. Inversely, the P fluxes found in Lampedusa* are the highest observed in the western basin, since P loss by dissolution induces a decrease of fluxes with the contribution of wet deposition. These results show that the wet deposition of dust in western Mediterranean Sea predominate on the fluxes of metals and P associated to intense dust deposition events. Thus, considering that one sample corresponds to one dust event (Vincent et al., 2016), the fluxes obtained for the site of the first category represent the mean trace metals mass flux for one intense dust deposition over western Mediterranean basin between 2011 and 2013. It is also the case for P mass fluxes for the sites of Mallorca and Corsica. Due to the effect of dissolution emphasized from aerosol analogues during the CARAGA protocol testing, these fluxes are underestimated at the worst case by 13 % for P and 10 % for trace metals. These mean fluxes are critical to estimate the global atmospheric inputs of trace metals and P since the most intense weekly dust deposition samples account in mass 50-84 % of the total deposition (Vincent et al., 2016). It has to be noted that the mass fluxes estimated in this paper are averaged weekly mass fluxes for the most intense dust events, as studied by Vincent et al. (2016): we could not compare with references values in literature which are typically annual.



## 5 Conclusion

The estimation of atmospheric deposition mass fluxes of nutrients and trace metals in the Western Mediterranean Sea is critical to understand the role of dust deposition on phytoplankton activity. Very few deposition measurements are available to document nutrients inputs for several years in the Western Mediterranean Sea. To estimate these nutrients and trace metals

mass flux, we validated the possibility to use samples mass measurement of atmospheric deposition performed in the same condition with CARAGA collectors at 5 sites in the western part of the Mediterranean. A first step was to estimate elemental mass loss during the in-situ collection and ignition protocol used for weighing samples. According to these tests with dust analogues and traffic soot, solubility larger than 15 % was observed for major metals, such as Ca, Mg and K, excluding the possibility to use the common Ca/Fe ratio to estimate source areas of deposited dust. However, the relative mass loss by

dissolution of 3 aerosol analogues was lower than 3 % for dust tracers like Ti, Al, Fe, and trace metals such as Cd, Co, Cr, Ni and V. The relative mass loss by dissolution was up to10 % for Zn, Cu and Mn. For the major nutrient P, this loss by dissolution ranged from 10 to 13 %. During the ignition, the elemental mass loss for tracer elements, nutrients and trace metals was not significant. These results point out that there is no bias due to the sampling method and the lab protocols used to estimate elemental fluxes from CARAGA atmospheric deposition samples.

Chemical analyses of 107 CARAGA samples corresponding to the most intense dust deposition events observed between 2011 and 2013 in the western basin were performed to validate using CARAGA samples to estimate origin of atmospheric deposition. After checking samples chemical composition using PCA, Al content and inter-elemental mass ratios X/Al, the chemical composition of samples from all the station excluding Frioul have typical chemical signature of Saharan dust, except for Zn in Lampedusa, Cr in Corsica and P in Le Casset and Lampedusa. So these CARAGA samples were chemically

exploitable and have been used to estimate P and trace metals mass fluxes associated to dust deposition keeping in mind the underestimate of soluble fraction.  The deposition sampled at Frioul could be affected by other sources even during an intense dust event, and hence elemental atmospheric fluxes measured at this site are linked to a mixing between dust and other relatively local sources.

The atmospheric elemental fluxes estimated from CARAGA samples constitute the first regional assessment of a mean weekly

mass fluxes of trace metals and P linked to intense dust deposition over western Mediterranean basin. The mean mass fluxes were dependent on the distance to the source and on the contribution of wet deposition. Knowing that a good representation of these intense fluxes is critical to estimate the role of atmospheric deposition on marine biosphere in modelling (Guieu et al., 2014), the parameterization of wet deposition of nutrients and trace metals needs to be improved.

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





Table 1: Limit of detection (LoD: limit of detection calculated with 3 times the standard deviation for 9 blanks), and elemental concentrations in blanks and CARAGA samples (in ppb).

| Elements | | LoD | Mean for blanks | Sample (2 mg) |
|---|---|---|---|---|
| **Marker elements** | Al | 2.2 | 2.5 ± 0.7 | 1395.5 |
| | Ti | 0.16 | 0.11 ± 0.05 | 73.76 |
| **Nutrients** | Fe | 0.4 | 0.9 ± 0.1 | 773.6 |
| | P | 6.2 | < LoD | 157.3 |
| **Trace metals** | Cd | 0.29 | 0.63 ± 0.07 | < LoD |
| | Co | 0.57 | 0.64 ± 0.19 | 8.67 |
| | Cr | 0.6 | < LoD | 34.6 |
| | Cu | 1.0 | < LoD | 16.4 |
| | Mn | 0.4 | 0.8 ± 0.1 | 71.5 |
| | Ni | 0.9 | 1.2 ± 0.3 | 18.3 |
| | V | 0.7 | 1.1 ± 0.2 | 22.1 |
| | Zn | 0.9 | < LoD | 36.0 |

Table 2: Elemental mass loss in soils of Douz and Banizoumbou, and traffic soot during the collection and ignition of wet deposition.

| Elements | | Soil of Douz | | Soil of Banizoumbou | | Traffic soot | |
|---|---|---|---|---|---|---|---|
| | | Mean (%) | SD (%) | Mean (%) | SD (%) | Mean (%) | SD (%) |
| **Markers** | Ca | 18.09 | 0.29 | 27.0 | 1.4 | 33.57 | 0.51 |
| | Al | 1.09 | 0.13 | 0.39 | 0.05 | 0.61 | 0.03 |
| | Mg | 3.43 | 0.03 | 30.40 | 0.51 | 8.81 | 0.18 |
| | Na | 10.09 | 0.56 | 4.60 | 1.27 | 95.5 | 2.2 |
| | K | 5.80 | 0.12 | 3.12 | 0.07 | 15.05 | 0.57 |
| | Ti | 0.57 | 0.09 | 0.12 | 0.02 | 8.55 | 2.89 |
| **Nutrients** | P | 13.3 | 2.3 | * | * | 10.3 | 1.0 |
| | Fe | 0.68 | 0.13 | 0.19 | 0.03 | 0.11 | 0.005 |





|  | | Cd | * | * | * | * | 4.97 | 0.78 |
|---|---|---|---|---|---|---|---|---|
| Trace metals | | Co | 1.12 | 0.10 | 0.19 | 0.03 | 2.41 | 0.27 |
| | | Cr | 1.01 | 0.11 | 0.11 | 0.02 | 0.38 | 0.01 |
| | | Cu | 3.87 | 0.35 | 1.67 | 0.46 | 6.58 | 0.12 |
| | | Mn | 2.78 | 0.06 | 10.77 | 0.20 | 7.52 | 0.08 |
| | | Ni | 0.62 | 0.07 | * | * | 4.67 | 0.17 |
| | | V | 3.47 | 0.12 | 0.38 | 0.06 | 2.00 | 0.09 |
| | | Zn | 4.77 | 0.62 | 6.39 | 0.86 | 8.75 | 0.10 |

*: Elemental concentration were under the limit of quantitation (LoQ), so we could not calculate the elemental loss.

Standard deviation (SD) results from triplicate experiments. Al, Ca, P, K, Na and Mg have been measured by ICP-AES, and other elements by HR-ICP-MS.

Table 3. PCA factor loadings for 10 elements and for 107 samples, F, factor.

| Sites | Le Casset | | | | Corsica | | | Mallorca | | Lampedusa | | | Frioul | | |
|---|---|---|---|---|---|---|---|---|---|---|---|---|---|---|---|
| Elements | F1 | F2 | F3 | F4 | F1 | F2 | F3 | F1 | F2 | F1 | F2 | F3 | F1 | F2 | F3 |
| **Al** | **0.97** | | | | **0.97** | | | **0.99** | | **0.70** | -0.70 | | **0.88** | | |
| **Fe** | **0.97** | | | | **0.99** | | | **0.99** | | **0.90** | 0.41 | | **0.98** | | |
| **P** | **0.77** | | | 0.54 | **0.94** | | | **0.99** | | | | **0.93** | **0.71** | -0.31 | -0.51 |
| **Ti** | **0.97** | | | | **0.99** | | | **0.99** | | **0.74** | -0.65 | | **0.95** | | |
| **Co** | 0.35 | **0.93** | | | **0.82** | | 0.54 | **0.86** | 0.51 | **0.81** | -0.40 | | **0.89** | -0.33 | |
| **Cr** | **0.98** | | | | 0.33 | **0.93** | | **0.99** | | **0.92** | | | **0.70** | 0.39 | 0.53 |
| **Cu** | **0.80** | | 0.46 | | **0.95** | | | **0.99** | | **0.74** | **0.60** | | **0.92** | | |
| **Mn** | **0.91** | | | | **0.96** | | | **0.99** | | **0.94** | | | **0.96** | | |
| **Ni** | **0.98** | | | | **0.91** | | | **0.94** | | **0.72** | 0.38 | **0.52** | **0.86** | | 0.44 |
| **V** | **0.98** | | | | **0.99** | | | **1.00** | | **0.77** | -0.62 | | **0.97** | | |
| **Zn** | **0.76** | | 0.49 | | **0.97** | | | **0.98** | | **0.59** | **0.71** | | **0.52** | **0.79** | -0.30 |
| Cumulative variances % | 76.8 | 85.5 | 91.5 | 95.5 | 83.1 | 93.2 | 96.3 | 94.9 | 97.7 | 57.3 | 82.2 | 94.1 | 66.9 | 84.5 | 93.0 |

Only loads larger than 0.3 (in absolute values) are reported. Loads larger than 0.6 are in bold.





Table 4: Al contents in deposition samples for 5 sites: Le Casset, Corsica, Mallorca, Lampedusa and Frioul

| Sites | Le Casset | Corsica | Mallorca | Lampedusa | Frioul |
|---|---|---|---|---|---|
| Mean (%) | 10.7 | 8.9 | 7.7 | 6.1 | 5.0 |
| ± 1 SD (%) | 4.1 | 1.5 | 3.4 | 2.1 | 2.4 |
| R$^2$ | 0.86 | 0.91 | 0.99 | 0.80 | 0.35 |

Table 5: Inter-elemental ratio in samples collected in the different CARAGA sites and reference values

| Sites | | Le Casset | Corsica | Mallorca | Lampedusa | Lampedusa* | Frioul | Rf1 | Rf2 | Rf3 |
|---|---|---|---|---|---|---|---|---|---|---|
| **Samples number** | | 17 | 11 | 21 | 37 | 32 | 21 | | | |
| **Fe/Al** | Mean | 0.54 | 0.64 | 0.57 | 1.09 | 0.66 | 0.55 | 0.56-0.65 | 0.50-0.78 | 0.40-1.69 |
| | ± 1 SD | 0.03 | 0.09 | 0.18 | 10.24 | 0.79 | 0.21 | | | |
| | R$^2$ | 0.99 | 0.95 | 1.00 | 0.12 | 0.98 | 0.73 | | | |
| **Ti/Al** | Mean | 0.06 | 0.07 | 0.06 | 0.08 | 0.07 | 0.06 | 0.06-0.21 | 0.07-0.13 | 0.06-0.11 |
| | ± 1 SD | 0.01 | 0.01 | 0.02 | 0.03 | 0.01 | 0.03 | | | |
| | R$^2$ | 0.98 | 0.96 | 0.99 | 0.99 | 1.00 | 0.61 | | | |
| **P/Al** | Mean | 0.03 | 0.01 | 0.01 | 0.05 | 0.03 | 0.14 | 0.010-0.018 | 0.01-0.02 | 0.007-0.012 |
| | ± 1 SD | 0.02 | 0.01 | 0.03 | 0.32 | 0.08 | 0.14 | | | |
| | R$^2$ | 0.52 | 0.84 | 0.97 | 0.01 | 0.45 | 0.33 | | | |
| **Co/Al 10$^{-3}$** | Mean | 0.65 | 0.74 | 0.70 | 0.69 | 064 | 0.22 | 0.48-0.82 | | |
| | ± 1 SD | 0.66 | 0.28 | 1.15 | 1.51 | 0.64 | 0.09 | | | |
| | R$^2$ | 0.07 | 0.64 | 0.64 | 0.63 | 0.71 | 0.73 | | | |
| **Cr/Al 10$^{-3}$** | Mean | 1.17 | 1.85 | 1.05 | 1.77 | 1.47 | 3.46 | 1.44-1.54 | | |
| | ± 1 SD | 0.26 | 3.14 | 1.03 | 7.67 | 2.05 | 2.59 | | | |
| | R$^2$ | 0.94 | 0.20 | 0.95 | 0.37 | 0.67 | 0.24 | | | |
| **Cu/Al 10$^{-3}$** | Mean | 0.87 | 0.65 | 0.47 | 1.15 | 0.61 | 5.66 | 0.43-0.50 | | 0.38 |
| | ± 1 SD | 0.57 | 0.30 | 1.31 | 18.21 | 1.22 | 13.18 | | | |





|  |  |  |  |  |  |  |  |  |  |
|---|---|---|---|---|---|---|---|---|---|
|  | $R^2$ | 0.49 | 0.86 | 0.92 | 0.01 | 0.70 | 0.01 |  |  |
| **Mn/Al** | Mean | 6.34 | 6.17 | 5.47 | 10.70 | 7.23 | 6.21 | 10.81-13.54 | 5.9 |
| **$10^{-3}$** | $\pm$ *1 SD* | *2.83* | *1.11* | *4.91* | *94.20* | *8.42* | *2.27* |  |  |
|  | $R^2$ | 0.67 | 0.90 | 0.99 | 0.26 | 0.94 | 0.70 |  |  |
| **Ni/Al $10^{-3}$** | Mean | 0.57 | 0.78 | 0.62 | 1.13 | 0.63 | 0.90 | 0.46-0.68 | 0.48 |
|  | $\pm$ *1 SD* | *0.18* | *0.51* | *1.24* | *9.42* | *0.58* | *0.40* |  |  |
|  | $R^2$ | 0.89 | 0.75 | 0.85 | 0.07 | 0.86 | 0.51 |  |  |
| **V/Al** | Mean | 1.55 | 1.68 | 1.25 | 1.53 | 1.47 | 1.34 | 2.15-2.84 | 1.40 |
| **$10^{-3}$** | $\pm$ *1 SD* | *0.26* | *0.32* | *0.66* | *2.95* | *0.37* | *0.42* |  |  |
|  | $R^2$ | 0.96 | 0.94 | 0.99 | 0.98 | 0.99 | 0.82 |  |  |
| **Zn/Al** | Mean | 2.64 | 2.09 | 1.55 | 122.63 | 14.52 | 15.86 | 1.13-1.50 | 1.01 1.70 |
| **$10^{-3}$** | $\pm$ *1 SD* | *1.56* | *0.91* | *2.85* | *1765.23* | *59.53* | *18.43* |  |  |
|  | $R^2$ | 0.44 | 0.89 | 0.91 | 0.01 | 0.001 | 0.12 |  |  |

*\*: Mean, standard deviation and correlation coefficient obtained after discarding 4 outlier samples for Lampedusa.*

*Rf1: reference values in soils of Banizoumbou and Douz, Rf2: reference values in wet deposition of dust near emitting area (Desboeufs et al.; 2010) or in total deposition with dust signature in the western Mediterranean basin (Ridame et al.; 1999; Guieu et al.; 2010)., Rf3: reference values in transported Saharan dust (Formenti et al., 2003, 2008, 2011 and 2014) for the Saharan emitting sources (PSA 1 to 3 : Northern Algeria / southern Tunisia, Mauritania / Morocco and northern Mali / southern Algeria respectively), corresponding with the identified provenance of our samples (Vincent et al.; 2016).*

Table 6: Mean weekly mass fluxes of nutrients and trace metals for 107 intense events samples, standard deviation (SD) and contribution of wet deposition on total fluxes for 5 studied sites

| **Sites** |  |  | **Le Casset** | **Corsica** | **Mallorca** | **Lampedusa\*** | **Lampedusa** | **Frioul** |
|---|---|---|---|---|---|---|---|---|
| **mg m$^{-2}$ wk$^{-1}$** | **Al** | Mean | 7.76 | 14.92 | 37.40 | 27.44 | 25.01 | 7.23 |
|  |  | *±1 SD* | *5.12* | *10.86* | *74.40* | *41.91* | *40.14* | *3.99* |
|  |  | Wet (%) | 71 | 88 | 86 | 57 | 55 | 71 |
|  | **Fe** | Mean | 4.16 | 9.48 | 21.46 | 17.93 | 27.30 | 3.97 |
|  |  | *±1 SD* | *2.65* | *7.21* | *41.15* | *23.10* | *47.08* | *2.77* |
|  |  | Wet (%) | 72 | 89 | 80 | 60 | 36 | 68 |



|  |  |  |  |  |  |  |  |
|---|---|---|---|---|---|---|---|
|  |  | Mean | 0.24 | 0.19 | 0.37 | 0.99 | 1.16 | 0.99 |
|  | **P** | ±1 SD | 0.16 | 0.13 | 0.54 | 1.41 | 1.64 | 1.16 |
|  |  | Wet (%) | 79 | 91 | 80 | 67 | 51 | 60 |
|  |  | Mean | 5.04 | 11.46 | 26.10 | 17.48 | 17.27 | 1.57 |
|  | **Co** | ±1 SD | 5.77 | 6.33 | 26.48 | 16.34 | 16.31 | 1.27 |
|  |  | Wet (%) | 80 | 90 | 69 | 49 | 44 | 75 |
|  |  | Mean | 9.05 | 33.03 | 39.11 | 39.80 | 44.23 | 25.02 |
|  | **Cr** | ±1 SD | 4.25 | 20.84 | 54.76 | 41.16 | 48.19 | 20.22 |
|  |  | Wet (%) | 69 | 87 | 79 | 57 | 46 | 77 |
|  |  | Mean | 6.73 | 10.14 | 17.43 | 16.84 | 28.88 | 40.94 |
|  | **Cu** | ±1 SD | 2.50 | 4.50 | 19.02 | 12.84 | 51.70 | 115.52 |
| $\mu g\ m^{-2}\ wk^{-1}$ |  | Wet (%) | 65 | 84 | 76 | 64 | 34 | 82 |
|  |  | Mean | 49.21 | 91.24 | 204.68 | 194.93 | 261.70 | 44.92 |
|  | **Mn** | ±1 SD | 31.68 | 79.30 | 349.24 | 273.03 | 421.64 | 26.65 |
|  |  | Wet (%) | 69 | 91 | 81 | 65 | 43 | 67 |
|  |  | Mean | 4.40 | 12.05 | 23.14 | 19.59 | 28.23 | 6.48 |
|  | **Ni** | ±1 SD | 2.14 | 5.79 | 28.49 | 21.15 | 38.19 | 3.90 |
|  |  | Wet (%) | 68 | 84 | 83 | 66 | 41 | 73 |
|  |  | Mean | 12.01 | 25.16 | 46.72 | 40.20 | 38.30 | 9.68 |
|  | **V** | ±1 SD | 6.56 | 17.26 | 75.20 | 58.24 | 55.39 | 6.62 |
|  |  | Wet (%) | 70 | 88 | 81 | 59 | 55 | 70 |
|  |  | Mean | 20.47 | 32.15 | 58.00 | 345.48 | 3067.19 | 114.69 |
|  | **Zn** | ±1 SD | 11.24 | 16.05 | 59.51 | 576.11 | 13063.10 | 153.65 |
|  |  | Wet (%) | 70 | 85 | 76 | 61 | 7 | 72 |

*: not including 5 outlier samples enriched in trace metals (especially Fe) in Lampedusa.



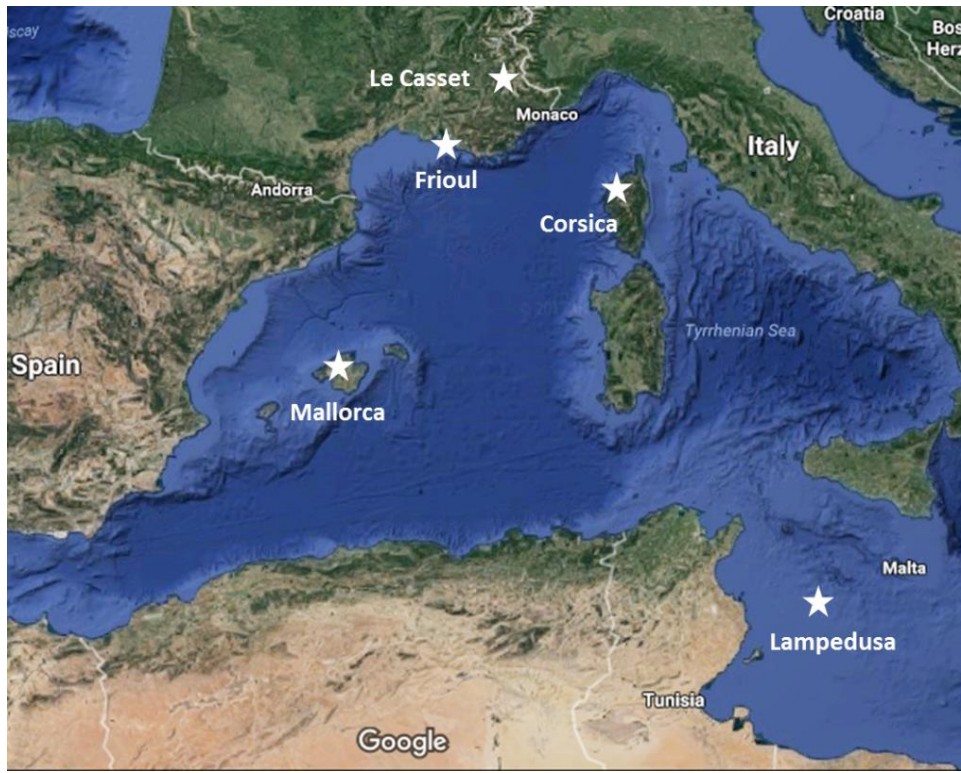

Figure 1: Location of the CARAGA sampling sites in the western Mediterranean basin: Le Casset, Corsica, Frioul, Mallorca and Lampedusa





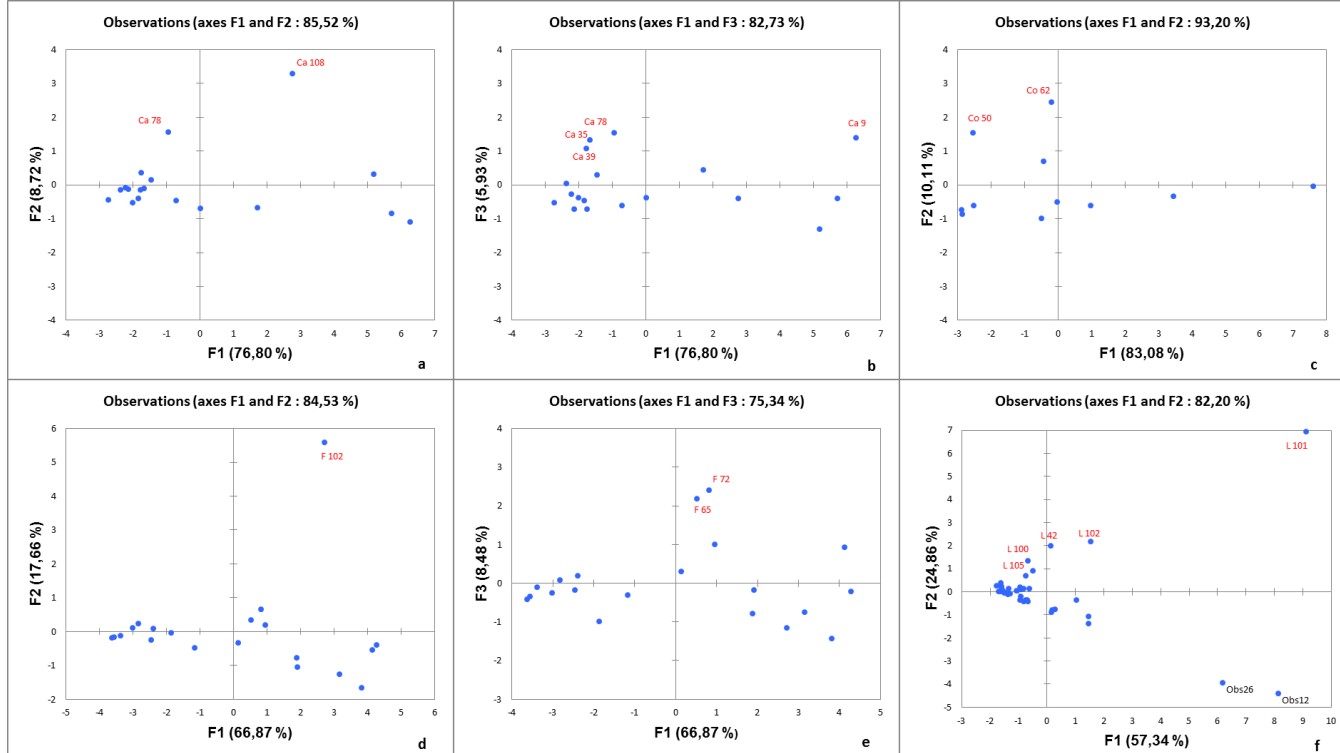

*a: Outlier samples of Co at Le Casset; b: Outlier samples of Zn and Cu at Le Casset; c: Outlier samples Cr Corsica; d: Outlier sample of Zn and Cu (and partially of Cr) at Frioul; e: Outlier samples of Cr at Frioul; f: Outlier samples of Zn, Cu and Fe at Lampedusa.*

Figure 2: Outlier samples identified by PCA for 4 sampling sites: Le Casset, Corsica, Frioul and Lampedusa.

