# Peer review of "Estimating chemical composition of atmospheric deposition fluxes from mineral insoluble particles deposition collected in the Western Mediterranean region"

_Atmospheric Measurement Techniques, 2017_

## Referee Comment (RC1) · Anonymous Referee #2 · 12 Jun 2017

Overall Comments: The manuscript estimates the insoluble chemical composition of bulk deposition samples obtained by using CARAGA sampler. Samples are collected at La Casset, Corsica and Frioul Island (France), Mallorca Island (Spain) and Lapedusa Island (Italy) situated at Western Mediterranean. Samples are analyzed for Al, CA, K, Mg, Na, Ti, P, Fe, Cd, Co, Cr, Cu, Mn, Ni, V and Zn applying Inductively Coupled Plasma-Atomic Emission Spectrometry (ICPAES). The manuscript evaluates the relative loss by dissolution in wet deposition in order to test the efficiency of the CARAGA sampler since it only collects insoluble fraction of bulk deposition. Furthermore, the

manuscript demonstrates chemical composition of in soluble bulk deposition at La Casset, Corsica and Frioul Island (France), Mallorca Island (Spain) and Lapedusa Island (Italy). In this respect, it may be the interest of scientific community. Consequently I suggest acceptance of the manuscript. However, before that the manuscript should be revised.

General Comments:

1-Title: Since the manuscript only focuses on the samples obtained from Western Mediterranean, Western Mediterranean should be used instead of Mediterranean.

2- Abstract: The references in the abstract should be removed from the text.

3- Experimental: Brief, information about sampling sites (such as site character, elevation and distance from major pollution sources) would be helpful for reader

4- Results and Discussion: Figure 1 is obtained by Excel and it has low resolution, better diagram should be presented. Results from the sites could be presented in a single diagram instead of Table 5 since it is difficult to follow crowded number. The difference between sampling sites should be supported by using t-test (if there is a normal distribution, if not then non-parametric t-test).

---

## Referee Comment (RC2) · Anonymous Referee #1 · 21 Jun 2017

Estimating chemical composition of atmospheric deposition fluxes from mineral insoluble particles deposition collected in the Mediterranean region. By Fu et al.

The paper aims to evaluate the metals and P deposition fluxes over the Mediterranean region by the chemical analysis of mineral atmospheric deposition sampled by CARAGA device located in 4 sites in the Mediterranean region. The data on deposition chemical characterization are scarce and the analytical work made by the authors is appreciable, therefore the paper deserves the publication on Atmospheric Measurement Techniques. Anyway, some corrections are necessary before the publication, here below some suggestions. In the data interpretation the authors approach is sometimes circular: they choose the main Saharan dust events deposition (demonstrated by the high load and beckward trajectories analysis), besides their sampler is able to capture the only mineral fraction of the deposition and they want demonstrate that the elements they measure in these samples are marker of dust deposition (see for instance sentence al page 9 lines 31-34). This conclusion is obvious for the main crustal markers (Al, Fe, Ti, Mn). Besides, the authors concludes that as these events cover the majority of deposition, the total contents of the metals arise from dust. I do not agree with such conclusion, this conclusion can be true only for the elements having only the crustal source (Al, Fe, Ti) accounting for a large amount to the total mass (Al, Fe), the fraction of metal deposited by Saharan dust for metals having also anthropic source can be quantified only by the analysis of all the samples not only those representing Saharan dust. Another weak point of the paper is the discussion of the elemental loss. An interesting analytical work is done to assess this loss for the metals, but the results are not well valorized. The importance of the soluble fraction of metals and especially P has to be highlighted. Here below some specific (and minor) comments that I hope can help the authors to improve the discussion.

Specific and minor comments

Page 3 lines 8-14. Sites description is very poor, the characterization of the different type of depositions needs to know more information about the aerosol source affecting the sampling sites. There is a reference to Vincent et al. 2016, but also in this paper only the geographical position of the sampling sites is reported.

Page 3 line 24. I suppose that "selected samples" has to be deleted, please check this sentence and correct it.

Section 3.1. The discussion in this section is reductive respect to the obtained results. I understand that the aim of this section is to demonstrate that you determine the total deposition flux and the "loss" are considered as a negative result, but, in my opinion,

the quantification of metals and nutrient solubility is a very important result and deserve a deep discussion. Besides, literature data on solubility in environmental condition are scarce. In this way I strongly suggest to change the aims of this section focusing on the importance of the soluble part (and their variability in the different aerosol types) in fertilization processes. The importance of metal solubility is also claimed by the authors at page 7 lines 28-30 to explain the north –south different Al percentage in the total deposited mass than the soluble fraction seems to be not negligible as the authors state in this section.

Section 3.2.1 PCA shows that all elements (excluding one in each site) are grouped in F1 representing the crustal source. This is expected due to the choice of samples, and the exclusion of anomalous samples do not change the general result. Figure 2 caption need to be revised and please increses the size of characters in the figure plots.

Section 3.2.2 Page 7 line 20-25 and related table 4. The percentage of Al respect to total mass of deposition is sometime (Le Casset and Corsica) higher than the percentage of Al in the average upper continental crust. Is it possible a loss of carbonate during the ash procedures?

Page 8 lines 25-30. The source of Zn from waste incineration is true in general over Mediterranean region, but not at Lampedusa, where Zn arises from manufacturing of non-ferrous material (largely use in extreme marine environment) as correctly state by the authors in the previous sentence.

Section 3.3 This section is the most interesting of the paper but need to be rewritten. Page 10 lines 7-10 such information is not inferable from table 6. Page 10 line 21. I suppose the sentence should be "…fluxes of metals and P associated to intense dry deposition events." Page 10 lines 21-29. The sentence is not clear. I do not understand what the authors want to demonstrate.

Conclusion and abstract are too general, some specific results have to be reported.

---

## Author Comment (AC1) · 10 Jul 2017

**Answers for discussion comments referee 1:**

Thanks too much for reading this article, and giving us so useful suggestions for revision, and I am sorry for my late answers. I cited your comments below and my answers are in red color following each question.

Estimating chemical composition of atmospheric deposition fluxes from mineral insoluble particles deposition collected in the Mediterranean region. By Fu et al. The paper aims to evaluate the metals and P deposition fluxes over the Mediterranean region by the chemical analysis of mineral atmospheric deposition sampled by CARAGA device located in 4 sites in the Mediterranean region. The data on deposition chemical characterization are scarce and the analytical work made by the authors is appreciable, therefore the paper deserves the publication on Atmospheric Measurement Techniques. Anyway, some corrections are necessary before the publication, here below some suggestions.

In the data interpretation the authors approach is sometimes circular: they choose the main Saharan dust events deposition (demonstrated by the high load and backward trajectories analysis), besides their sampler is able to capture the only mineral fraction of the deposition and they want demonstrate that the elements they measure in these samples are marker of dust deposition (see for instance sentence on page 9 lines 31-34). This conclusion is obvious for the main crustal markers (Al, Fe, Ti, Mn).

We do not think our interpretation of data is circular. Even if the collector is limited to mineral deposition measurements, for trace metals it's not obvious that deposition is mainly due to dust but can be associated to anthropogenic particles. The use of "intense dust events" purposes to identify, even in the very constrained dusty cases, this anthropogenic influence. We use precisely the fact that it's no doubt during these events about the origin of main dust markers to identify other influence for trace metals by comparison with these markers. The conclusion mentioned in the sentence p9 lines 31-34 is only for trace metals, not for dust marker: "the results showed that the atmospheric deposition of trace metals at Le Casset, Corsica, Mallorca and Lampedusa is mainly associated to dust fluxes except Zn in Lampedusa and Cr in Corsica during intense dust event".

Besides, the authors concludes that as these events cover the majority of deposition, the total contents of the metals arise from dust. I do not agree with such conclusion, this conclusion can be true only for the elements having only the crustal source (Al, Fe, Ti) accounting for a large amount to the total mass (Al, Fe), the fraction of metal deposited by Saharan dust for metals having also anthropic source can be quantified only by the analysis of all the samples not only those representing Saharan dust.

We agree with this comment. Our conclusions are for the total dust deposition, but it was probably not so clear. So in order to clarify this point, we added P11 lines 28-30: Chemical analyses of 107 CARAGA samples corresponding to the most intense dust deposition events observed between 2011 and 2013 in the western basin were performed to validate using CARAGA samples to estimate  atmospheric deposition from dust.

Another weak point of the paper is the discussion of the elemental loss. An interesting analytical work is done to assess this loss for the metals, but the results are not well valorized. The importance of the soluble fraction of metals and especially P has to be highlighted.

The purpose of the paper is not a process study about dissolution. The experiment did in this work purpose to mimic a wet deposition then a rinsing in the CARAGA samples, we cannot hightlight the result on P soluble fraction which corresponds with a very specific condition of dissolution (see detailed response in specific comments).

Here below some specific (and minor) comments that I hope can help the authors to improve the discussion.

**Specific and minor comments:**

Page 3 lines 8-14. Sites description is very poor, the characterization of the different type of depositions needs to know more information about the aerosol source affecting the sampling sites. There is a reference to Vincent et al. 2016, but also in this paper only the geographical position of the sampling sites is reported.

We added information and references about aerosol influence on the stations in the chapter "2.1 Sampling of insoluble deposition and total mass measurements".

Page 3 line 24. I suppose that "selected samples" has to be deleted, please check this sentence and correct it.

The sentence has been corrected: Saharan origin was identified for 107 selected samples by using satellite data and air mass trajectories.

Section 3.1. The discussion in this section is reductive respect to the obtained results. I understand that the aim of this section is to demonstrate that you determine the total deposition flux and the "loss" are considered as a negative result, but, in my opinion, the quantification of metals and nutrient solubility is a very important result and deserve a deep discussion. Besides, literature data on solubility in environmental condition are scarce. In this way I strongly suggest to change the aims of this section focusing on the importance of the soluble part (and their variability in the different aerosol types) in fertilization processes. The importance of metal solubility is also claimed by the authors at page 7 lines 28-30 to explain the north –south different Al percentage in the total deposited mass than the soluble fraction seems to be not negligible as the authors state in this section.

We agree with the importance of the soluble part for studying the bioavailability of trace metals for phytoplankton. However, the methodology developed in our work was to estimate the loss of metals during the protocol of sampling and ignition. Thus, the solubility obtained in these tests is not representative of dissolution processes in rain or sea waters. Moreover, it's known that solubility of trace metals from dust is low (e.g. Desboeufs et al., 2005), it isn't a "new" finding. Concerning the decreasing of Al percentage, the soluble part of metals is not mentioned, only the soluble species as Ca are pointed (p8, lines 5-6): "This increase was explained by the fact that the mass loss due to dissolution of highest soluble species during settling modifies the mass percentage of the less soluble elements'

Section 3.2.1 PCA shows that all elements (excluding one in each site) are grouped in F1 representing the crustal source. This is expected due to the choice of samples, and the exclusion of anomalous samples do not change the general result. Figure 2 caption need to be revised and please increses the size of characters in the figure plots.

We agree that such PCA result on common factor for the majority of elements was expected because of the choice of samples. However, the objective in this approach was not to show the

crustal origin of all the elements whatever the samples, but precisely to identify the anomalous samples in order to discriminate the potential other sources. Actually, after removing outlier samples, the obtained fluxes for several elements were changed, e.g. Zn in Lampedusa.

Figure 2 has been corrected.

Section 3.2.2 Page 7 line 20-25 and related table 4. The percentage of Al respect to total mass of deposition is sometime (Le Casset and Corsica) higher than the percentage of Al in the average upper continental crust. Is it possible a loss of carbonate during the ash procedures?

The decomposition of $CaCO_3$ is from 800 °C, but the ignition protocol is just at 550 °C, so we don't think there was a loss of carbonate during ignition. The method of LOI (loss on ignition) in 550°C is traditionally used to determine SOM (soil organic matter), and just the loss of SOM and structural water for several minerals were reported, according to Sun et al. (2009).The higher Al contents are due to the loss of the most soluble species such as Ca carbonate or sulfate as detailed in p8 between lines 4 and 19.

Page 8 lines 25-30. The source of Zn from waste incineration is true in general over Mediterranean region, but not at Lampedusa, where Zn arises from manufacturing of non-ferrous material (largely use in extreme marine environment) as correctly stated by the authors in the previous sentence.

We agree that Lampedusa Island is not a metropolitan area, but there is a Power plant close to the sampling site (added on page 8 line 28), we are waiting for more information about this power plant form other colleagues, it will be added in article while possible. But we can say that the abundance of Zn in Lampedusa samples showed anthropogenic Zn sources affecting this sampling site.

Section 3.3: This section is the most interesting of the paper but need to be rewritten. Page 10 lines 7-10 such information is not inferable from table 6. Page 10 line 21. I suppose the sentence should be "… fluxes of metals and P associated to intense dry deposition events." Page 10 lines 21-29. The sentence is not clear. I do not understand what the authors want to demonstrate.

1) 2 categories added in table 6: dust deposition and mixed deposition.

2) The sentence has been corrected: "These results show that the fluxes of metals and P associated to wet deposition predominate in western Mediterranean Sea environment".

3) Page 10 lines 21-29: two points were developed: firstly, the mass fluxes estimated in this paper are averaged weekly mass fluxes for the most intense dust events account in mass 50-

84 % of the total deposition, as studied by Vincent et al. (2016), so we could not compare with references values in literature which are typically annual. Secondly, due to the effect of dissolution during the CARAGA sampling protocol, these fluxes are underestimated at the worst case by 13 % for P and 10 % for Zn, Cu and Mn, and 5% for other trace metals.

Conclusion and abstract are too general, some specific results have to be reported.

In abstract sentence added: 'the mass fluxes strongly depend on the distance from dust sources and the most intense events, proximity from anthropogenic sources strongly impacted the masse fluxes of Zn and Cu at Lampedusa and Frioul'

In conclusion sentences added : 'High average and strong standard deviation of Al (dust marker) mass fluxes were observed at Lampedusa and Mallorca due to several most intense dust events, and extreme Zn mass fluxes (122.63 ± 1765.23 µg m$^{-2}$ wk$^{-1}$) was observed at Lampedusa because of contamination of the incinerator sources'.

---

## Author Comment (AC2) · 10 Jul 2017

Dear,

Thanks too much for reading this article, and giving us so useful suggestions for revision, and I am sorry for my late answers, but I think I have already revised accroding your comments before the discussion version.

Best kinds

[Figure]

Fu Yinghe

---

## Referee Comment (RC3) · Anonymous Referee #1 · 20 Jul 2017

I'm surprised that authors do not capture the sense of my comments. With a very few changes they have the possibility to better highlight the results and give more relevance to the paper. Besides they insist on the attribution of Zn source from waste incinerator or power plant, there aren't any proofs for this in the paper and the related sentences have to be smoothed. In the .pdf you can find my point to point replay to the author answer.

Please also note the supplement to this comment:

[Figure]

https://www.atmos-meas-tech-discuss.net/amt-2017-100/amt-2017-100-RC3-supplement.pdf

[Figure]

**Supplement:**

Replay to the authors answer to the manuscript:

Estimating chemical composition of atmospheric deposition fluxes from mineral insoluble particles deposition collected in the Mediterranean region.

By Fu et al.

My reply to the authors answer is in bold character.

....

*In the data interpretation the authors approach is sometimes circular: they choose the main Saharan dust events deposition (demonstrated by the high load and backward trajectories analysis), besides their sampler is able to capture the only mineral fraction of the deposition and they want demonstrate that the elements they measure in these samples are marker of dust deposition (see for instance sentence on page 9 lines 31-34). This conclusion is obvious for the main crustal markers (Al, Fe, Ti, Mn).*

*We do not think our interpretation of data is circular. Even if the collector is limited to mineral deposition measurements, for trace metals it's not obvious that deposition is mainly due to dust but can be associated to anthropogenic particles. The use of "intense dust events" purposes to identify, even in the very constrained dusty cases, this anthropogenic influence. We use precisely the fact that it's no doubt during these events about the origin of main dust markers to identify other influence for trace metals by comparison with these markers. The conclusion mentioned in the sentence p9 lines 31-34 is only for trace metals, not for dust marker: "the results showed that the atmospheric deposition of trace metals at Le Casset, Corsica, Mallorca and Lampedusa is mainly associated to dust fluxes except Zn in Lampedusa and Cr in Corsica during intense dust event".*

**If the aim of the paper is find the influence of anthropic source of trace metal during Saharan dust events, the approach is correct but what is the relevance of this finding? In my opinion is more interesting to find the anthropic impact of trace metals to the total deposition. Anyway, the approach is correct, please change some sentences in order to be more clear on the aims of the paper.**

*Besides, the authors conclude that as these events cover the majority of deposition, the total contents of the metals arise from dust. I do not agree with such conclusion, this conclusion can be true only for the elements having only the crustal source (Al, Fe, Ti) accounting for a large amount to the total mass (Al, Fe), the fraction of metal deposited by Saharan dust for metals having also anthropic source can be quantified only by the analysis of all the samples not only those representing Saharan dust.*

*We agree with this comment. Our conclusions are for the total dust deposition, but it was probably not so clear. So in order to clarify this point, we added P11 lines 28-30: Chemical analyses of 107 CARAGA samples corresponding to the most intense dust deposition events observed between 2011 and 2013 in the western basin were performed to validate using CARAGA samples to estimate the origin of atmospheric deposition from dust.*

**The added sentence "estimate the origin of atmospheric deposition from dust", is still circular, if the deposition is from dust you already know the origin, it is dust.**

*Another weak point of the paper is the discussion of the elemental loss. An interesting analytical work is done to assess this loss for the metals, but the results are not well valorised. The importance of the soluble fraction of metals and especially P has to be highlighted.*

*The purpose of the paper is not a process study about dissolution. The experiment did in this work purpose to mimic a wet deposition then a rinsing in the CARAGA samples, we cannot highlight the result on P soluble fraction which corresponds with a very specific condition of dissolution (see detailed response in specific comments).*

**If the "*P soluble fraction corresponds with a very specific condition of dissolution*" and it is not representative of the real situation all the section is merely speculative.**

*Here below some specific (and minor) comments that I hope can help the authors to improve the discussion.*
*Specific and minor comments:*
*Page 3 lines 8-14. Sites description is very poor, the characterization of the different type of depositions needs to know more information about the aerosol source affecting the sampling sites. There is a reference to Vincent et al. 2016, but also in this paper only the geographical position of the sampling sites is reported.*

*We added information and references about aerosol influence on the stations in the chapter "2.1 Sampling of insoluble deposition and total mass measurements".*

I'll wait for the new version to see the new section 2.1.

*Page 3 line 24. I suppose that "selected samples" has to be deleted, please check this sentence and correct it.*
*The sentence has been corrected: Saharan origin was identified for 107 selected samples by using satellite data and air mass trajectories.*

**Ok**

*Section 3.1. The discussion in this section is reductive respect to the obtained results. I understand that the aim of this section is to demonstrate that you determine the total deposition flux and the "loss" are considered as a negative result, but, in my opinion, the quantification of metals and nutrient solubility is a very important result and deserve a deep discussion. Besides, literature data on solubility in environmental condition are scarce. In this way I strongly suggest to change the aims of this section focusing on the importance of the soluble part (and their variability in the different aerosol types) in fertilization processes. The importance of metal solubility is also claimed by the authors at page 7 lines 28-30 to explain the north –south different Al percentage in the total deposited mass than the soluble fraction seems to be not negligible as the authors state in this section.*

*We agree with the importance of the soluble part for studying the bioavailability of trace metals for phytoplankton. However, the methodology developed in our work was to estimate the loss of metals during the protocol of sampling and ignition. Thus, the solubility obtained in these tests is not representative of dissolution processes in rain or sea waters. Moreover, it's known that solubility of trace metals from dust is low (e.g. Desboeufs et al., 2005), it isn't a "new" finding.*

**I'm not asking for a big change of this section, just few sentences to highlight the importance of the soluble part, but if the Authors declare that "the solubility obtained in these tests is not representative of dissolution processes in rain" and especially during the rain episode sampled by CARAGA, the section is merely speculative and have to be deleted.**

*Concerning the decreasing of Al percentage, the soluble part of metals is not mentioned, only the soluble species as Ca are pointed (p8, lines 5-6): "This increase was explained by the fact that the mass*

*loss due to dissolution of highest soluble species during settling modifies the mass percentage of the less soluble elements'*

**Ok**

*Section 3.2.1 PCA shows that all elements (excluding one in each site) are grouped in F1 representing the crustal source. This is expected due to the choice of samples, and the exclusion of anomalous samples do not change the general result. Figure 2 caption need to be revised and please increses the size of characters in the figure plots.*

*We agree that such PCA result on common factor for the majority of elements was expected because of the choice of samples. However, the objective in this approach was not to show the crustal origin of all the elements whatever the samples, but precisely to identify the anomalous samples in order to discriminate the potential other sources. Actually, after removing outlier samples, the obtained fluxes for several elements were changed, e.g. Zn in Lampedusa.*
*Figure 2 has been corrected.*

**This is not a common use of PCA but could be right**

*Section 3.2.2 Page 7 line 20-25 and related table 4. The percentage of Al respect to total mass of deposition is sometime (Le Casset and Corsica) higher than the percentage of Al in the average upper continental crust. Is it possible a loss of carbonate during the ash procedures?*
*The decomposition of $CaCO_3$ is from 800 °C, but the ignition protocol is just at 550 °C, so we don't think there was a loss of carbonate during ignition. The method of LOI (loss on ignition) in 550°C is traditionally used to determine SOM (soil organic matter), and just the loss of SOM and structural water for several minerals were reported, according to Sun et al. (2009).The higher Al contents are due to the loss of the most soluble species such as Ca carbonate or sulfate as detailed in p8 between lines 4 and 19.*

**Ok**

*Page 8 lines 25-30. The source of Zn from waste incineration is true in general over Mediterranean region, but not at Lampedusa, where Zn arises from manufacturing of non-ferrous material (largely use in extreme marine environment) as correctly stated by the authors in the previous sentence.*
*We agree that Lampedusa Island is not a metropolitan area, but there is a Power plant close to the sampling site (added on page 8 line 28), we are waiting for more information about this power plant form other colleagues, it will be added in article while possible. But we can say that the abundance of Zn in Lampedusa samples showed anthropogenic Zn sources affecting this sampling site.*

**It is well known that both waste incineration and power plant do not emit only Zn, but also other metals and you do not observe other anomalous enrichment of other metals, especially if you consider samples characterises by high Saharan dust content. I'm still not convinced about the Zn source from waste incineration or power plant.**

*Section 3.3: This section is the most interesting of the paper but need to be rewritten. Page 10 lines 7-10 such information is not inferable from table 6. Page 10 line 21. I suppose the sentence should be "... fluxes of metals and P associated to intense dry deposition events." Page 10 lines 21-29. The sentence is not clear. I do not understand what the authors want to demonstrate.*

*1) 2 categories added in table 6: dust deposition and mixed deposition.*

*2) The sentence has been corrected: "These results show that the fluxes of metals and P associated to wet deposition predominate in western Mediterranean Sea environment".*

*3) Page 10 lines 21-29: two points were developed: firstly, the mass fluxes estimated in this paper are averaged weekly mass fluxes for the most intense dust events account in mass 50-84 % of the total*

*deposition, as studied by Vincent et al. (2016), so we could not compare with references values in literature which are typically annual. Secondly, due to the effect of dissolution during the CARAGA sampling protocol, these fluxes are underestimated at the worst case by 13 % for P and 10 % for Zn, Cu and Mn, and 5% for other trace metals.*

**Ok these results have to be shortly reported in the conclusion.**

*Conclusion and abstract are too general, some specific results have to be reported.*

*In abstract sentence added: 'the mass fluxes strongly depend on the distance from dust sources and the most intense events, proximity from anthropogenic sources strongly impacted the masse fluxes of Zn and Cu at Lampedusa and Frioul'*

**Ok**

*In conclusion sentences added : 'High average and strong standard deviation of Al (dust marker) mass fluxes were observed at Lampedusa and Mallorca due to several most intense dust events, and extreme Zn mass fluxes (122.63 ± 1765.23 µg m$_{-2}$ wk$_{-1}$) was observed at Lampedusa because of contamination of the incinerator sources'.*

**Too many digit in the Zn mass fluxes.**
**I strongly disagree with the incinerator source for Zn, the sentence have to be changed in: "…and extreme Zn mass fluxes (123 ± 1765 µg m-2 wk-1) was observed at Lampedusa that need to be deeply investigate but likely due to the large use of non-ferrous metal manufacturing in this environment strongly impacted by sea spray."**

84 % of the total deposition, as studied by Vincent et al. (2016), so we could not compare with references values in literature which are typically annual. Secondly, due to the effect of dissolution during the CARAGA sampling protocol, these fluxes are underestimated at the worst case by 13 % for P and 10 % for Zn, Cu and Mn, and 5% for other trace metals.

Conclusion and abstract are too general, some specific results have to be reported.
In abstract sentence added: 'the mass fluxes strongly depend on the distance from dust sources and the most intense events, proximity from anthropogenic sources strongly impacted the masse fluxes of Zn and Cu at Lampedusa and Frioul'
In conclusion sentences added : 'High average and strong standard deviation of Al (dust marker) mass fluxes were observed at Lampedusa and Mallorca due to several most intense dust events, and extreme Zn mass fluxes (122.63 ± 1765.23 µg m$_{-2}$ wk$_{-1}$) was observed at Lampedusa because of contamination of the incinerator sources'.

---

## Author Response (AR1)

Original questions are cited, and my answer is in blue.

*1) In the data interpretation the authors approach is sometimes circular: they choose the main Saharan dust events deposition (demonstrated by the high load and backward trajectories analysis), besides their sampler is able to capture the only mineral fraction of the deposition and they want demonstrate that the elements they measure in these samples are marker of dust deposition (see for instance sentence on page 9 lines 31-34). This conclusion is obvious for the main crustal markers (Al, Fe, Ti, Mn).*

*We do not think our interpretation of data is circular. Even if the collector is limited to mineral deposition measurements, for trace metals it's not obvious that deposition is mainly due to dust but can be associated to anthropogenic particles. The use of "intense dust events" purposes to identify, even in the very constrained dusty cases, this anthropogenic influence. We use precisely the fact that it's no doubt during these events about the origin of main dust markers to identify other influence for trace metals by comparison with these markers. The conclusion mentioned in the sentence p9 lines 31-34 is only for trace metals, not for dust marker: "the results showed that the atmospheric deposition of trace metals at Le Casset, Corsica, Mallorca and Lampedusa is mainly associated to dust fluxes except Zn in Lampedusa and Cr in Corsica during intense dust event".*

**If the aim of the paper is find the influence of anthropic source of trace metal during Saharan dust events, the approach is correct but what is the relevance of this finding? In my opinion is more interesting to find the anthropic impact of trace metals to the total deposition. Anyway, the approach is correct, please change some sentences in order to be clearer on the aims of the paper.**

**We agree that it will be more interesting to find the anthropogenic impact of trace metals to the total deposition, but to do that, we have to analyze all samples, not only the most intense dust deposition samples. So in this paper (with 107 samples) which aims to assess and validate the suing of CARAGA system to estimate the mass fluxes of nutrients and trace metals, we can just characterize these samples, and try to get more information from these samples.**

**In the introduction and conclusion, we changed some sentences to clarify the aims of this paper:**

**1: in introduction page 3 lines 4-5: In the present study, we investigate the possibility of using the CARAGA ashed samples to estimate the elemental mass deposition flux of nutrients and trace metals.**

**2: in conclusion page 11 lines 28-30 : Chemical analyses of 107 CARAGA samples corresponding to the most intense dust deposition events observed between 2011 and 2013 in the western basin were performed to validate using CARAGA samples to estimate the elemental mass deposition flux of nutrients and trace metals.**

*2) Besides, the authors conclude that as these events cover the majority of deposition, the total contents of the metals arise from dust. I do not agree with such conclusion, this conclusion can be true only for the elements having only the crustal source (Al, Fe, Ti) accounting for a large amount to the total mass*

*(Al, Fe), the fraction of metal deposited by Saharan dust for metals having also anthropic source can be quantified only by the analysis of all the samples not only those representing Saharan dust.*

*We agree with this comment. Our conclusions are for the total dust deposition, but it was probably not so clear. So in order to clarify this point, we added P11 lines 28-30: Chemical analyses of 107 CARAGA samples corresponding to the most intense dust deposition events observed between 2011 and 2013 in the western basin were performed to validate using CARAGA samples to estimate the origin of atmospheric deposition from dust.*

**The added sentence "estimate the origin of atmospheric deposition from dust", is still circular, if the deposition is from dust you already know the origin, it is dust.**

*The sentence should be the following:*

**Chemical analyses of 107 CARAGA samples corresponding to the most intense dust deposition events observed between 2011 and 2013 in the western basin were performed to validate using CARAGA samples to estimate the elemental mass deposition flux of nutrients and trace metals.**

*Another weak point of the paper is the discussion of the elemental loss. An interesting analytical work is done to assess this loss for the metals, but the results are not well valorised. The importance of the soluble fraction of metals and especially P has to be highlighted.*

**The underestimate of soluble part of trace metals and P were highlighted in 3 parts. Details in the explication in 4).**

*The purpose of the paper is not a process study about dissolution. The experiment did in this work purpose to mimic a wet deposition then a rinsing in the CARAGA samples, we cannot highlight the result on P soluble fraction which corresponds with a very specific condition of dissolution (see detailed response in specific comments).*

If the "*P soluble fraction corresponds with a very specific condition of dissolution*" and it is not representative of the real situation all the section is merely speculative.

Explication is the same as in 4).

*3) Here below some specific (and minor) comments that I hope can help the authors to improve the discussion.*

*Specific and minor comments:*

*Page 3 lines 8-14. Sites description is very poor, the characterization of the different type of depositions needs to know more information about the aerosol source affecting the sampling sites. There is a reference to Vincent et al. 2016, but also in this paper only the geographical position of the sampling sites is reported.*

*We added information and references about aerosol influence on the stations in the chapter "2.1 Sampling of insoluble deposition and total mass measurements".*

I'll wait for the new version to see the new section 2.1.

Section added:

2.1 Sampling of insoluble deposition and total mass measurements

Weekly deposition samples were collected between 2011 and 2013 with CARAGA collectors at 5 stations in the western Mediterranean basin presented in Fig. 1. The sites positions were selected to cover the western basin by integrating East-West and North-South gradients:

1. Le Casset (44°59 N, 6°28 E, S-E France at 1850 m in altitude, rural area, ~ 170 Km from sea shore). A previous study about composition of rainwaters (Coddeville et al., 2002) showed that this site was very influenced by the anthropogenic emissions and the crustal sources around the Mediterranean Basin and North Africa. Other European areas (e.g., Italy) can influence the concentrations recorded at Le Casset.

2. Corsica Island (43°00 N, 9°21 E, France, at 75 m in altitude, 300 m from sea shore). The chemical composition of aerosol on this site was studied during the ChArMEx campaign (June-August 2013) and the results showed the influence of elemental carbon-containing particles, issued in part from fossil fuel, biomass burning and ship traffic, sea-salt particles, and secondary organic aerosol (Mallet et al., 2016; Arndt et al., 2017).

**3. Frioul Island (43°15 N, 5°17 E, France, at 45 m in altitude, in front of Marseille (city of 1. 57 million population according to INSEE 2012). Previous dry deposition measurements operated in the Frioul Islands have shown, from lipid analyses, a significant contribution of terrestrial particles, plant debris, and meat cooking residues in the deposited particles (Rontani et al., 2012).**

**4. Mallorca Island (39°15 N, 3°03 E, Spain, at 7 m in altitude, 70 m from sea shore). This site had never been used for atmospheric studies, so no information about the source influence is available.**

**5. Lampedusa Island (33°21 N, 10°30 E, Italy, at 45 m in altitude, ~ 20 m from the north-western coast). The site on Lampedusa Islands is a station for climate observations, maintained by ENEA (the Italian Agency for New Technologies, Energy, and Sustainable Economic Development). Various atmospheric parameters are monitored on this station: greenhouse gases concentrations, aerosol chemical and optical properties, total ozone, radiative budget, The results show that the PM10 concentrations are influenced by marine and ship emissions to which are added Saharan dust events in spring (Becagli et al., 2012, Mallet et al., 2016). About trace metals concentrations, Ni and V in the submicronic fraction are mainly due to heavy oil combustion associated to ship traffic (Becagli et al., 2012 and 2015).**

*4) Section 3.1. The discussion in this section is reductive respect to the obtained results. I understand that the aim of this section is to demonstrate that you determine the total deposition flux and the "loss" are considered as a negative result, but, in my opinion, the quantification of metals and nutrient solubility is a very important result and deserve a deep discussion. Besides, literature data on solubility in environmental condition are scarce. In this way I strongly suggest to change the aims of this section focusing on the importance of the soluble part (and their variability in the different aerosol types) in fertilization processes. The importance of metal solubility is also claimed by the authors at page 7 lines 28-30 to explain the north –south different Al percentage in the total deposited mass than the soluble fraction seems to be not negligible as the authors state in this section.*

*We agree with the importance of the soluble part for studying the bioavailability of trace metals for phytoplankton. However, the methodology developed in our work was to estimate the loss of metals during the protocol of sampling and ignition. Thus, the solubility obtained in these tests is not representative of dissolution processes in rain or sea waters. Moreover, it's known that solubility of trace metals from dust is low (e.g. Desboeufs et al., 2005), it isn't a "new" finding.*

**I'm not asking for a big change of this section, just few sentences to highlight the importance of the soluble part, but if the Authors declare that "the solubility obtained in these tests is not representative of dissolution processes in rain" and especially during the rain episode sampled by CARAGA, the section is merely speculative and have to be deleted.**

**The underestimate of soluble part of trace metals and P were highlighted in 3 parts:**

**In the end of sections 3.1:  To conclude, taking into account the underestimate of soluble fraction, the CARAGA samples seem to be relevant for estimating the total deposition fluxes of Al, Fe, P, Co, Cr, Cu, Ni, Mn, Ti, V and Zn.**

**In the end of section 3.3:  Secondly, due to the effect of dissolution during the CARAGA sampling protocol, these fluxes are underestimated at the worst case by 13 % for P and 10 % for Zn, Cu and Mn, and 5% for other trace metals.**

**In the conclusion: So these CARAGA samples were chemically exploitable and have been used to estimate P and trace metals mass fluxes associated to dust deposition keeping in mind the underestimate of soluble fraction.**

**The solubility is specific for this deposition collection system: rain + rinse (250 ml artificial rainwater and 100ml of rinse water), so the loss by dissolution is the summation of 2 times dissolution: rain process in situ, and rinse process in lab to collect the deposition. But the fractional solubility studied in the literature is based in one process: the first one. That's why we said "the solubility obtained in these tests is not representative of dissolution processes in rain".**

5) *Page 8 lines 25-30. The source of Zn from waste incineration is true in general over Mediterranean region, but not at Lampedusa, where Zn arises from manufacturing of non-ferrous material (largely use in extreme marine environment) as correctly stated by the authors in the previous sentence.*

*We agree that Lampedusa Island is not a metropolitan area, but there is a Power plant close to the sampling site (added on page 8 line 28), we are waiting for more information about this power plant form other colleagues, it will be added in article while possible. But we can say that the abundance of Zn in Lampedusa samples showed anthropogenic Zn sources affecting this sampling site.*

**It is well known that both waste incineration and power plant do not emit only Zn, but also other metals and you do not observe other anomalous enrichment of other metals, especially if you consider samples characterises by high Saharan dust content. I'm still not convinced about the Zn source from waste incineration or power plant.**

**We agree with you, this power plant is not the major anthropogenic Zn sources. the sentence about power plant deleted.**

*In conclusion sentences added : 'High average and strong standard deviation of Al (dust marker) mass fluxes were observed at Lampedusa and Mallorca due to several most intense dust events, and extreme Zn mass fluxes (122.63 ± 1765.23 µg m$_{-2}$ wk$_{-1}$) was observed at Lampedusa because of contamination of the incinerator sources'.*

Too many digit in the Zn mass fluxes.

I strongly disagree with the incinerator source for Zn, the sentence have to be changed in: "…and extreme Zn mass fluxes (123 ± 1765 µg m-2 wk-1) was observed at Lampedusa that need to be deeply investigate but likely due to the large use of non-ferrous metal manufacturing in this environment strongly impacted by sea spray."

About the anthropogenic Zn, I agree with you, this power plant is not the major anthropogenic Zn sources. And the sentence has been changed as you suggested.

[revised manuscript text omitted]